# Effect of Fresh Citrus Pulp Supplementation on Animal Performance and Meat Quality of Feedlot Steers

**DOI:** 10.3390/ani11123338

**Published:** 2021-11-23

**Authors:** Santiago Luzardo, Georgget Banchero, Virginia Ferrari, Facundo Ibáñez, Gonzalo Roig, Valentín Aznárez, Juan Clariget, Alejandro La Manna

**Affiliations:** 1Programa de Producción de Carne y Lana y Plataforma Agroalimentos, Instituto Nacional de Investigación Agropecuaria (INIA), Estación Experimental INIA Tacuarembó, Ruta 5 km 386, Tacuarembó 45000, Uruguay; 2Programa de Producción de Carne y Lana, Instituto Nacional de Investigación Agropecuaria (INIA), Estación Experimental INIA La Estanzuela, Ruta 50 km 11, Colonia 70000, Uruguay; gbanchero@inia.org.uy (G.B.); jclariget@inia.org.uy (J.C.); alamanna@inia.org.uy (A.L.M.); 3Plataforma Agroalimentos, Instituto Nacional de Investigación Agropecuaria (INIA), Estación Experimental W. Ferreira Aldunate, Ruta 48 km 10, Canelones 90100, Uruguay; vferrari@inia.org.uy (V.F.); fibanez@inia.org.uy (F.I.); 4MARFRIG Group, Ruta 2 km 288, Río Negro 65000, Uruguay; gonzalo.roig@marfrig.com (G.R.); valentin.aznarez@marfrig.com (V.A.)

**Keywords:** citrus pulp, steers, animal performance, meat quality, antioxidant

## Abstract

**Simple Summary:**

The use of crop, fruit, and vegetable processing co-products for animal feeding has been of increasing interest worldwide to minimize feed waste. Additionally, it has a positive impact from an environmental standpoint being a more efficient use of feed resources. However, the use of many co-products has limitations related to poor palatability for animals and the logistical aspects of product delivery. Citrus pulp is a by-product of the citrus industry presenting a nutritional composition that makes it attractive for use as an ingredient in animal feeding. Previous research has shown that it is feasible to utilize citrus pulp in beef cattle rations. The objective of the present study was to evaluate the effect of the inclusion of fresh citrus pulp in the diet of feedlot steers on animal performance and carcass and meat nutritional properties and quality. The findings of the study showed that fresh citrus pulp may be used as an energy source in rations for feedlot steers as it does not affect animal performance or carcass and meat quality, but, rather, has a positive effect on dry matter intake and a better feed to gain ratio.

**Abstract:**

The use of fruit by-products such as citrus pulp represents a feeding ingredient that deserves to be evaluated as an energy source in animal rations. Thirty-six British breed steers were allotted to one of the three feeding treatments (12 steers/treatment): 0%, 15% and 30% of fresh citrus pulp inclusion in the ration in a randomized complete block design to evaluate animal performance and carcass and meat quality traits. In the present study, the inclusion of fresh citrus pulp up to 30% of the diet did not affect the animal average daily gain (*p* > 0.05) but steers that were fed the pulp consumed less feed (*p* < 0.05) and presented a lower feed conversion ratio (*p* < 0.05) than their counterparts without citrus pulp in their diet. No effect of fresh citrus pulp was observed on carcass and meat quality (*p* > 0.05). A greater lipophilic antioxidant capacity (*p* < 0.05) in meat was observed when fresh citrus pulp was offered at 15% of the diet. Fresh citrus pulp used up to 30% as a feed ingredient in feedlot rations does not negatively affect animal performance or meat quality but, rather, has a positive effect on dry matter intake and a better feed conversion ratio.

## 1. Introduction

In recent decades, there has been an increasing interest for the reutilization of fruit and vegetable processing co-products in farm animal nutrition due to the social and environmental pressures faced by modern society [1,2,3,4]. In addition, the use of fruit waste for animal feeding would reduce feeding costs incurred by farmers [5]. Citrus pulp represents an important by-product for the Uruguayan citrus juice industry that can be used as a high energy feed in ruminant rations [6]. Residues of citrus juice production are principally composed of water, soluble sugars, fiber, organic acids, amino acids, proteins, minerals, and lipids, as well as flavonoids and vitamins [7]. The performance of steers fed citrus pulp in their ration has been similar to those fed with corn diets [8] being suitable for its inclusion in a balanced diet replacing other energy feeds [9]. The inclusion of citrus pulp into a corn-silage diet has shown no significant changes in the acetic to propionic acid ratio in the rumen and although not considered a roughage, the pulp might contain roughage-like properties that tend to promote higher ruminal pH values [6].

On the other hand, using plant by-products containing phytochemicals, such as citrus pulp, can enhance the deposition of bioactive compounds in muscle tissues that delay the oxidative deterioration of color and flavor, and extend the shelf-life of meat [10]. Limited data exist on the effect of citrus pulp on beef meat quality although several studies have been performed in lambs [11,12,13]. Feeding whole citrus pulp to lambs enhances the antioxidant status of muscle more through an increase in the deposition of α-tocopherol than through the effect of flavonoids [14].

The hypothesis of the present study was that the inclusion of fresh citrus pulp up to 30% of the ration (dry-matter basis) of steers fed high-concentrate diets could be used as a feed ingredient lowering feeding costs. In lambs, greater levels of citrus pulp of 30% have shown negative impacts such as rumen parakeratosis [6].

The objective of the investigation was to evaluate the effect of increasing levels of fresh citrus pulp on steers fed high-concentrate diets, on voluntary dry matter intake, animal performance, ruminal pH, meat quality traits and its antioxidant capacity.

## 2. Materials and Methods

The experiment was carried out at the Intensive Beef Fattening Unit of Marfrig Group “El Impulso” (33°12′ S and 58°05′ W), Rio Negro, Uruguay. The feeding period was of 104 d starting on 6 August and ending on 18 November 2020. All methods and conditions employed in this study were approved by the Committee for the Ethical Use of Animals of the National Institute Agricultural Research, Uruguay (INIA, Protocol number 2020-12).

### 2.1. Experimental Design and Dietary Treatments

Thirty-six Angus, Hereford, and Angus–Hereford cross steers were blocked by weight and breed and were assigned randomly to one of the three dietary treatments. At the start of the experiment, steers weighed 384 ± 26.4 kg. The steers of the first treatment were offered a control diet without fresh citrus pulp (FCP0), the steers of the second treatment were offered the same diet as those in treatment 1, but replacing corn silage and corn grain with 15% of fresh citrus pulp (FCP15) on a dry-matter basis and the steers of the third treatment were offered the same diet as those in treatment 2, but with 30% of fresh citrus pulp (FCP30) on dry-matter basis (Table 1). The three levels of FCP were defined to study the response curve to FCP inclusion in the diet. The greater proportion of FCP (FCP30) was defined based on previous research [6] where they reported that high levels of citrus pulp in the diet could have negative impacts on animal performance. Fresh citrus pulp was mainly composed of lemon with an energy concentration of 2.54 Mcal/kg DM of ME and 6.94% of CP (DM basis).

The experiment was a randomized complete block design with three treatments and 12 replications in which each animal was placed in an individual pen. The experimental pens had a mean size of 50 m^2^ (2.5 m × 20 m) divided with two strands of electric wire, fed by a solar-panel electrifier. Each pen had its own drinking and feeding trough which were half plastic tanks with a capacity of 100 L. Pens were ballast soil surfaced. Ballast soil consists of compacted broken stone and is used to give stability to the surface to avoid mud.

The feed was offered twice per day, and the same diet was offered throughout the experimental period. The first meal at 08:00 h provided 60% of the diet, and the second meal at 14:00 h provided the remaining 40% of the diet. The adaptation period was of 14 d, starting on August 6 and ending on August 20. The different components of the diets were weighed separately with a digital scale and then mixed and offered to each animal.

Diet adjustments were based on slick-bunk management [15,16] and weighed rejects. Diet adjustment was performed every day based on observation of the bunk before the first meal was offered in the morning, with a scale from 1 to 5 (1 = slick bunk, 5 = full bunk). One animal of the FCP30 treatment was removed by day 50 of the experimental period because it refused to consume the diet although no health problems were detected.

### 2.2. Animal Determinations

At the beginning of the trial, the steers were drenched for fluke (Fasimac^®^, Elanco, Bogotá, Colombia) and vaccinated for respiratory diseases (HBV1, Pi3, *Pasteurella multocida*, *Histophilus somni* and *Arcanobacterium pyogenes*) with Alliance Respiratoria^®^ (Boehringer Ingelheim, Buenos Aires, Argentina), and for clostridial diseases with Sintoxan^®^ (Merial, Buenos Aires, Argentina).

Daily dry matter intake (DMI, dry-matter basis) was calculated from the difference between the quantity offered and refused each day multiplied by its DM percentage. Steers were weighed without fasting at 08:00 h every 14 days before the first meal was provided. Average daily gain (ADG) was calculated as a linear regression of each weight of each individual animal [17]. Feed to gain ratio (F:G) was calculated by dividing the average daily DMI by the average daily gain. Water intake (WI) was calculated as the difference between the remaining water level and the volume needed to reach its original level in the tank. Rumen pH was measured hourly in 12 animals (four animals per treatment in the same block) using ruminal boluses (SmaXtec Premium Bolus, SmaXtec Animal Care GmbH, Graz, Austria) throughout the experimental period. The boluses were administered by a veterinarian from the company country representative, using an oral applicator. Boluses were powered by a lithium battery, which communicated through an internal antenna with an external receiver.

### 2.3. Feed Determinations

Diets were sampled weekly and monthly composites were sent to the Nutrition Laboratory of INIA La Estanzuela to perform the chemical composition (Table 1). Dry matter content of the diets was determined by drying them in a forced-air oven at 60 °C for 48 h according to Harris [18]. The DM content (%) was calculated using (dry weight/wet weight) × 100. The neutral detergent fiber (NDF) and the acid detergent fiber (ADF) concentrations corrected by ash were determined using an Ankom fiber analyzer (ANKOM-2000I; ANKOM Technology, Macedon, NY, USA) [19]. Ash was determined by the method 942.05 of AOAC [20]. Ether extract was determined using ANKOM-XT15 (ANKOM Technology, Macedon, NY, USA) according to the AOAC method 954.02 [20], and acid detergent lignin (ADL) was determined as described by Goering and Van Soest [21]. Nitrogen (N) was measured using a N analyzer (Kjeltec 8200; FOSS Analytical, Hillerod, Denmark), and crude protein (CP) was calculated as N × 6.25.

### 2.4. Energy Calculation of the Diet

Samples of each individual component of the diet were chemically analyzed and total digestible nutrients (TDN) were calculated according to Weiss et al. [22]. Conversion from TDN to net energy of maintenance (NEm) or net energy of gain (NEg) was performed using calculation formulas of the NASEM [23].

### 2.5. Carcass and Meat Quality Determinations

Steers were slaughtered in a commercial meat processing plant where hot carcass weight (HCW) was recorded after slaughter. After 48 h chilling, carcasses were ribbed between the 10th and 11th rib and one hour later marbling score (MARB), ribeye area (REA), and fat thickness (FAT) were measured. Marbling scores were determined using the USDA degrees of marbling [24] and subcutaneous fat thickness was measured over the ribeye between the 10th and 11th ribs in the fourth quartile from the chine along the dorsal side of the Longissimus thoracis (LT) muscle. Ribeye area was traced onto acetate tracing paper and later the area was measured with a software (Foxit Software, Fremont, CA, USA). Instrumental meat color (CIE L*: lightness, a*: redness and b*: yellowness) was also measured on each left half-carcass side in triplicate with a Minolta colorimeter CR-400 (Konica Minolta Sensing Inc., Osaka, Japan) using a C illuminant, a 2° standard observer angle, and an 8 mm aperture size, and calibrated with a white tile before use.

At the deboning room, a 5 cm sample was removed from the LT muscle (from the 11th rib following caudal direction) of each left half-carcass. The samples were vacuum-packaged and transported to the Meat Laboratory of INIA Tacuarembó. Each meat sample was divided into two steaks of 2.5 cm thickness. In one steak, subcutaneous fat was removed and cut into small pieces to be subsequently frozen at −80 °C. It was later pulverized using a Robot Coupe R2 (Robot Coupe^®^, Montceau-les-Mines, France). Immediately following homogenization, each sample was packed in individual sterile whirl-pack bags (Nasco, Fort Atkinson, WI, USA) and placed into a −80 °C freezer until micronutrients and antioxidant capacity determinations were performed.

The other steak was aged at 0–2 °C for 5 days. After aging, instrumental lean color (CIE L*: lightness, a*: redness and b*: yellowness) was measured on each steak in triplicate after 40 min blooming with a colorimeter (as previously described). Subsequently, Warner-Bratzler shear force (WBSF; model D2000- WB, G-R Electric Manufacturing Co, Manhattan, KS, USA) was assessed according to the American Meat Science Association guidelines [25]. Steaks were weighed before and after cooking and cooking losses were calculated as ((weight of raw steak − weight of cooked steak)/weight of raw steak) × 100. Steaks were cooked in a preheated clam-shell-style grill (GRP100 The Next Grilleration, Spectrum Brands, Inc., Miami, FL, USA) until the internal temperature reached 71 °C. After cooking, six cores of 1.27 cm diameter were removed from each steak parallel to the longitudinal orientation of muscle fibers. Individual shear force values were averaged to assign a mean peak WBSF value to each sample. Steaks were weighed before and after cooking and the cooking losses were calculated as a percentage of the weight of the raw meat.

### 2.6. Micronutrient Analysis

The extraction procedures for micronutrients and antioxidant capacity determinations were performed in raw LT lyophilized samples. Tocopherol, retinoid, and carotenoid derivatives were analyzed following the procedures described by Xu [26] and Bertolín et al. [27], with modifications. Briefly, 0.1 g of dry samples was vortexed, sonicated (10 min, 20 °C) and extracted overnight at room temperature with 1 mL 10% potassium hydroxide (ethanol:water, 50:50). After the addition of 1 mL of 10 μg/mL butylated hydroxytoluene in hexane:ethyl acetate (9:1), the tubes were vortexed for 15 min and centrifuged at 12,500 rpm for 5 min. The extraction procedure was repeated twice and the upper layers were collected and evaporated under vacuum in a CentriVap™ concentrator (Labconco Co., Kansas City, MO, USA) at 30 °C for 30 min. The dry residue was dissolved in 0.5 mL of methanol, vortexed for 1 min and filtered (0.22 μm) into an amber vial. Finally, 50 μL were injected on a HPLC Prominence Modular System (Shimadzu Co., Kyoto, Japan) equipped with a Nucleosil C18 column (5 μm; 250 × 4.6 mm), a diode array detector (PDA, SPD-M20A), and a fluorescence (RF-10AXL, Shimadzu, Torrance, CA, USA) detectors, controlled by the LabSolution Software^®^ (Shimadzu, Torrance, CA, USA). The mobile phase, with a flow rate of 1.5 mL.min^−1^, consisted of acetonitrile (solvent A), methanol (solvent B), and water (solvent C). The HPLC flow gradient started with isocratic 60:35:5 (A:B:C) up to 5 min, 2 min to 60:40:0, 3 min at 60:40:0, 10 min to 22:78:0, 5 min at 22:78:0, and then 10 min to initial condition. The temperature of the autosampler and the column were adjusted at 15 °C and 35 °C, respectively. Alpha tocopherol was detected by fluorescence set at 295 nm for excitation wavelength and 330 nm for the emission wavelength. Retinol and derivatives were determined by UV-vis absorbance at 325 nm and lutein, β-carotene, and carotenoids at 450 nm. The analytes were identified by comparison of the retention times and spectral analysis with those of the pure standards and data reported in literature. Data analysis was performed by applying LabSolution Shimadzu Software^®^ (Shimadzu, Torrance, CA, USA). Alpha tocopherol, retinoid, and carotenoid concentrations were expressed as μg α-tocopherol/g, μg retinol/g, and μg β-carotene/g of muscle (wet basis), respectively.

### 2.7. Antioxidant Capacity Determinations

For 1,1-diphenyl-2-picrylhydrazyl free radical (DPPH) and hydrophilic oxygen radical absorbance capacity (ORAC hydrophilic) assays, 0.1 g of lyophilized samples was mixed with 1 mL of methanol:water (80:20) solution, homogenized, and placed in an ultrasonic bath for 10 min at 20 °C, and then stored for 4 h at 20 °C avoiding light exposure. After that, samples were centrifuged at 12,500 rpm for 5 min to obtain a methanolic extract. All spectrophotometric data were obtained using a multi-mode microplate reader Synergy H1^®^, with two automatic reagent dispensers (BioTek Instruments Inc., Winooski, VT, USA). The quantification of free-radical-scavenging of DPPH (2,2-diphenyl-1-picrylhydrazyl) was performed mixing 15 μL of appropriated diluted extract with 235 μL DPPH solution (125 mM). The mixture was shaken and incubated for 24 h at 20 °C in dark conditions. The reduction of light absorption was measured at 517 nm and quantified using a Trolox standard curve. The results were expressed as μmol Trolox equivalents (TE) per 100 g of muscle (wet basis).

The ORAC assay was performed based on the methodology described by Wu et al. [28] with some modifications. For the hydrophilic ORAC assay, an aliquot (25 μL) of the diluted methanolic extracts or control (gallic acid) or standard solutions of Trolox (0 to 60 mM) was transferred to 96-well microplates. In the lipophilic ORAC assay 0.1 g of dry samples was mixed with 2.0 mL of hexane, vortexed, and sonicated for 10 min at 20 °C. Then, the hexane layer was removed and evaporated for 20 min at 30 °C in the vacuum CentriVap concentrator. The residue was dissolved in 50 μL of acetone and then diluted with 150 μL of 7% MCD solution (methyl-β-cyclodextrin, in acetone:water 50:50). The MCD solution (7%) was used for sample dilution, as for Trolox standards (0, 10, 20, 30, 40, 50, and 60 mM). Both ORAC analyses followed the same kinetic procedure. Each plate was placed in the microplate reader and 150 μL of fluorescein solution (0.008 μM) prepared with 75 mM phosphate buffer was added using the automatic dispenser. After shaking and incubation at 37 °C for 30 min, 25 μL AAPH (153 mM) freshly prepared with 75 mM phosphate buffer was added to each well. The plate was shaken, and the fluorescence was monitored every minute for 60 min. The fluorescence wavelengths were set at 485 nm for excitation and 528 nm for emission. The results were estimated based on the standard curve of Trolox concentrations and the areas under the fluorescence decay curves using Biotek Gen 5™ version 3.09.07 (BioTek Instruments, Inc., Winooski, VT, USA) coupled to the microplate reader. The ORAC activity was expressed as μmol Trolox equivalents (TE) per 100 g of muscle (wet basis).

### 2.8. Statistical Analysis

The experiment was analyzed as a randomized complete block design (RCBD) with feeding treatment (FCP0, FCP15, FCP30) as a fixed effect and the block was included as random using the PROC MIXED procedure of the SAS (SAS Institute, Cary, NC, USA, version 9.4). Ruminal pH was analyzed as repeated measures over time and the autoregressive (AR (1)) covariance structure was used based on the smallest value of the Akaike’s information criterion when compared to compound symmetry and unstructured covariances structures. Characteristics measured on the carcass were analyzed using covariates. Slaughter weight (SW) was used as a covariate to analyze HCW, and HCW was used as a covariate to analyze MARB, REA, and FAT. Homogeneity of variance and normality for all data was evaluated using the studentized residuals plots. Kenward–Roger approximation was used to calculate denominator degrees of freedom for different covariance structures for adjustment of the F-statistic. After ANOVA, least squares means were calculated for treatment comparisons with a significance level of α = 0.05, using the PDIFF option of LSMEANS, when F-tests were significant (*p* < 0.05). Orthogonal polynomial contrasts were used to determine the linear and quadratic effects of increasing citrus pulp levels on the response variables with a significance level of α = 0.05. Contrast of fresh citrus pulp (FCP15 + FCP30) treatments vs. FCP0 was also performed.

## 3. Results

### 3.1. Animal Performance

Steers did not differ (*p* > 0.05) among treatments on initial and final weight or on ADG (Table 2). Although animal performance was not affected by the inclusion of FCP in the diet, steers from FCP30 consumed less feed (*p* < 0.05) and were more efficient (*p* < 0.05) at converting feed nutrients into increased body mass than those animals from FCP0 and FCP15. Increasing levels of FCP in the diet linearly decreased (*p* < 0.05) dry matter intake and F:G. Water intake among treatments was similar, however, steers’ water intake per kg of DM consumed increased linearly (*p* < 0.05) as the level of FCP in the diet increased.

### 3.2. Ruminal pH

Ruminal pH values increased linearly (*p* < 0.05) as the level of FCP increased in the diet from 6:00 to 8:00 h. The first meal was offered to the steers at 8:00 h while the second was offered at 14:00 h. Subsequently, 3 h after the first meal (i.e., from 11:00 h) there was a quadratic effect (*p* < 0.05) of increased FCP levels for 12 consecutive hours (Figure 1). Furthermore, from 11:00 to 13:000 and from 15:00 to 17:00 h the ruminal pH of the steers from FCP15 and FCP30 treatments was greater (*p* < 0.05) than those from FCP0.

### 3.3. Carcass and Meat Quality

Traits measured on the carcass such as HCW, MARB, REA, or FAT were not affected (*p* > 0.05) by dietary treatments (Table 3). There were no significant differences (*p* > 0.05) among treatments on WBSF, cooking losses, or instrumental lean color.

When we analyzed the relationship between the response variables and the dietary treatments through orthogonal polynomial contrasts, linear and quadratic trends were not significant (*p* > 0.05) and neither were the contrast between FCP15 or FCP30 and FCP0 (without citrus pulp) for any of the variables.

### 3.4. Micronutrients and Meat Antioxidant Capacity

Meat from steers fed with FCP30 presented greater (*p* < 0.05) content of α-tocopherol compared to the other two treatments. In addition, meat from FCP15 had a greater (*p* < 0.05) α-tocopherol content than FCP0 (Table 4). No differences (*p* > 0.05) were observed in retinol content among dietary treatments although FCP0 showed a lower (*p* < 0.05) retinoid content than FCP30. It is important to note that α-tocopherol and retinoid contents increased linearly (*p* < 0.05) as the inclusion level of FCP in the diet increased. A greater (*p* < 0.05) content of β-carotene was observed on FCP15 compared to FCP30 although no differences (*p* > 0.05) were found on total carotenoid content among treatments.

Regarding antioxidant activity determinations, no differences (*p* > 0.05) were found in DPPH or hydrophilic ORAC values. Nevertheless, a greater (*p* < 0.05) lipophilic ORAC value was observed on meat from FCP15 than the other two treatments. It is worth noting that lipophilic ORAC values showed a quadratic response (*p* < 0.05) as the levels of FCP into the diet increased. Furthermore, we observed a greater (*p* < 0.05) lipophilic ORAC value in meat from treatments that included FCP (FCP15 or FCP30) in the diet compared to FCP0.

## 4. Discussion

Citrus pulp was included in the ration to formulate similar diets which would explain why no differences were observed in ADG among treatments. It is important to note that in our experiment steam flake was kept at the same level through the feeding treatments and corn silage and ground corn were the feed components replaced by FCP. Lenehan et al. [29] evaluated the effects of replacing rolled barley with citrus pulp and no differences were observed on ADG between dietary treatments of young growing cattle offered grass silage ad libitum. However, Gouvea et al. [30] reported an increase in ADG in Nellore bulls when they replaced ground corn—that has less energetic concentration than steam flake—with pelleted citrus pulp. Therefore, if the inclusion of FCP in the diet does not change the energetic concentration of the diet it would be expected to have no effect on ADG.

Regarding lower DMI and the better F:G of the steers with increasing levels FCP in the diet, we speculate that the rate of passage might slow down and thus feed digestibility could be increased to some extent compensating for lower FI. In our study, no differences were observed on WI among treatments despite that the inclusion of FCP that could meet part of water requirements of ruminants [6]. Nevertheless, the ratio WI:DMI was greater with increasing FCP levels which could be explained by the high bulk density of citrus pulp decreasing the DMI due to rumen fill [6].

Even though large amounts of pectin are present in citrus pulp that are rapidly and extensively degraded in the rumen, they yield little lactate causing less decline of ruminal pH [6]. The diets in our experiment were not particularly high in energy concentration and pH was never below 6.6 although differences were observed among treatments. Indeed, there was a quadratic effect of increased FCP levels after 3 h of the first meal and for 12 consecutive hours. It has been reported that there is no effect on ruminal pH when substituting levels of ground corn with pelleted citrus pulp [30]. On the other hand, previous research has found a reduction of ruminal pH in Jersey steers when dried citrus pulp in the ration reached to 67 and 100% replacement of corn silage [31]. In addition, Lenehan et al. [29] reported a decrease of ruminal pH when substituting barley (pH = 6.79) with citrus pulp (pH = 6.64). In our experiment, the reduction of corn grain and corn silage levels also induced a lower pH in the FCP30 treatment. However, pH values recorded hourly were far above the limit where subacute acidosis would occur (pH = 5.6) [32,33].

As with ADG, carcass traits such as HCW, REA, FAT, and MARB did not differ among treatments. These findings agree with previous research where no differences were observed in carcass traits of Nellore bulls when they compared four different dietary energy sources including citrus pulp [34]. In another study, researchers did not observe differences in carcass characteristics of heifers when evaluating two levels of ruminal undegradable protein and two levels of citrus pulp in the ration [35].

Oxidation of meat lipid and protein fractions has been reported as the main non-microbial cause of its quality deterioration during aging [36]. Tenderness is the most important trait affecting beef palatability and hence consumer acceptance [37,38,39]. In our study, WBSF values of meat aged for 5 d did not differ among feeding treatments which agrees with previous research [34,35].

Meat color is the most important characteristic that determines a consumer’s purchase decision [40]. Lipid and myoglobin oxidation processes in meat seem to be linked and the latter and are associated with lean discoloration due to the conversion of oxymyoglobin to metmyoglobin [41]. The concentration and activity of many endogenous skeletal muscle antioxidants can be influenced by diet [42,43]. An effect of dietary citrus pulp on the delay of lipid oxidation in lamb meat has been reported, but its effect on color stability has been unclear [12]. In the present study, instrumental color values (L*, a*, b*) of meat aged for 0 d and 5 d were not affected by the inclusion of FCP in the diet probably because the inclusion levels were lower than those required to affect meat color. Previous studies evaluating instrumental meat color (L*, a*, b*) in beef cattle observed greater L* values [44] or no differences [45] due to the inclusion of citrus pulp in the diet.

Salami et al. [45] studied the dietary effect of dried citrus pulp on α-tocopherol content on meat and they reported no differences among feeding treatments in which barley was replaced by 0%, 40%, and 80% of dried citrus pulp. On the other hand, it was observed that feeding lambs with a diet containing 35% of dried citrus pulp resulted in a threefold greater concentration of α-tocopherol in muscle [14]. In addition, it has been reported that meat from Angus steers that were fed with a dried citrus pulp (150 g/kg DM) diet for 90 days presented a threefold greater concentration of α-tocopherol than the control diet [44]. Our results showed that meat from FCP30 treatment had three times more α-tocopherol content than FCP0 achieving the suggested threshold concentration (3 to 3.5 µg/g muscle) to delay lipid and pigment oxidation [46,47,48]. Profile analyses for identification of other tocopherols and tocotrienols in further investigations could also contribute to corroborating its antioxidant bioactivity. Retinol content observed agrees with the levels previously reported [49], but its concentration did not differ among treatments. According to the extraction and analysis conditions, this retinoid would correspond to retinyl palmitate, although it is not possible to validate the identification. Previous studies in muscle tissue demonstrated that after absorption, retinoids are stored mainly as retinyl palmitate [50]. Other pro-vitamin-A compounds such as carotenoids and β-carotenes follow the same metabolic pathways. Research conducted in beef cattle showed that β-carotene is essentially the only carotenoid absorbed from the intestine [51]. It has been reported that β-carotene plays a role as an antioxidant under low accumulation of reactive substances, but its function may change to pro-oxidant, or it may be degraded, under higher accumulation [52]. Research has shown that β-carotene levels found in beef muscle are very dependable on the diet composition [51,53,54]. It has been reported that there is a wide range of β-carotene concentrations from 0.06 µg/g in steers under grain-based diets [54] to 2.00 µg/g in bulls fed with corn silage supplemented with enriched n-3 concentrate [52]. In our experiment, β-carotene concentrations in all treatments were within the mentioned range but closer to the highest concentration. A possible explanation might be that corn silage, which contains greater levels of β-carotene than forage hay, was used as a feed ingredient [55]. In addition, the FCP used was of lemon which has a greater concentration of β-carotene than orange pulp [56]. We hypothesize that a better combination of corn silage and FCP would explain the greater levels of β-carotene in FCP15 than in FCP30 treatment although no differences were observed in the carotenoid contents among treatments. Unfortunately, we were unable to know the concentrations of β-carotene and carotenoids in the diet. More research would be needed to understand the effect of the citrus pulp as a feed ingredient including its carotenoid profile and the mechanisms involved in its deposition in muscle tissue.

Values of the three antioxidant activity analyses were in the range of those reported in a previous study, considering that, in our experiment, we presented the values per 100 g of muscle [28]. A greater value of hydrophilic ORAC was found in FCP0 than in FCP15 or FCP30 treatments. Phenolic compounds and amino acids, particularly sulfur-containing amino acids, dipeptides such as carnosine and anserine, and antioxidant enzyme activity, among others, are responsible for the hydrophilic antioxidant activity [57,58]. On the other hand, antioxidants such as tocopherols, retinoids, and carotenoids, influence the lipophilic ORAC antioxidant activity and the FCP15 treatment showed a greater deposition of lipophilic antioxidant compounds into animal muscle. Beef is a complex matrix and there does not exist a single method that can fully detect its antioxidant status [28].

The inclusion of FCP into feeding rations would provide feasible opportunities to enhance the antioxidant activity and nutritional quality of beef although considerable variation in the chemical composition of the diet and citrus pulp can affect its effect.

## 5. Conclusions

We conclude that FCP may be used as a feed ingredient in rations for feedlot steers, not affecting animal performance but with a positive effect on DMI and better F:G. Carcass and meat quality were not affected by dietary treatments although the content of some micronutrients and the antioxidant capacity was increased when FCP was added in the diet. However, it is necessary to resolve some logistical aspects associated to storage and delivery of FCP to include this by-product as an animal feed ingredient.

## Figures and Tables

**Figure 1 animals-11-03338-f001:**
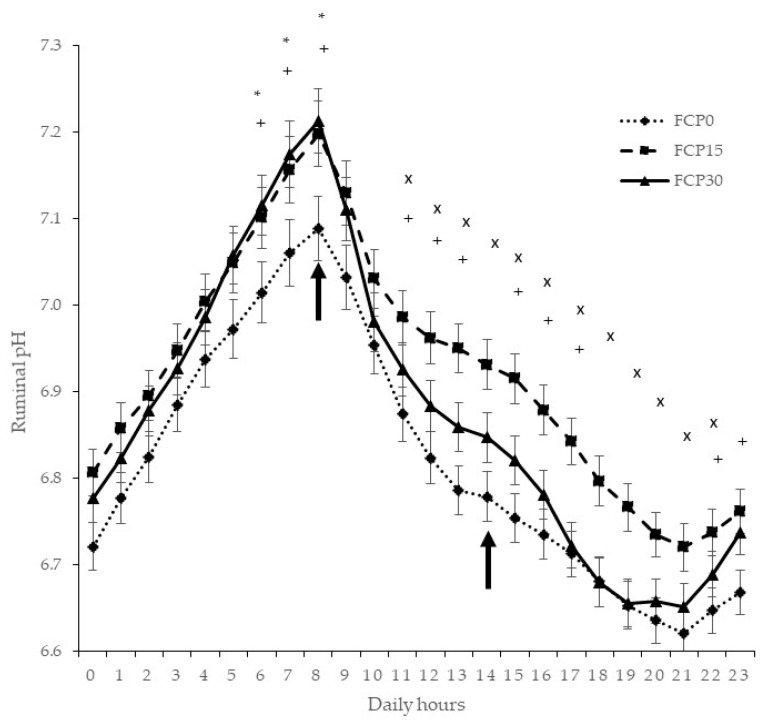
Least square means for hourly ruminal pH throughout the day by feeding treatment. Symbols indicate significant differences (*p* < 0.05). *: a linear contrast, x: a quadratic contrast among treatments, +: contrasts between FCP0 and FCP15 + FCP30 treatments. Black arrows indicate the time of feeding.

**Table 1 animals-11-03338-t001:** Feed ingredients and chemical composition of the diets.

Item	Dietary Treatments (Citrus Pulp as a Percentage of DM)		
	FCP0	FCP15	FCP30
Fresh citrus pulp (%)	0.0	15.0	30.0
Steam flake corn (%)	19.0	19.0	19.0
Corn grain (%)	13.3	5.8	0.0
Soybean meal (%)	11.8	11.8	11.8
Corn silage (%)	40.0	31.1	20.6
Wheat straw (%)	13.9	15.1	16.3
Premix (%)	1.3	1.3	1.3
Urea (%)	0.67	0.83	0.98
Premix ^1^ (%)	1.3	1.3	1.3
Physically effective ^2^ NDF (%)	22.8	22.1	21.0
Chemical Composition			
DM (% as fed)	51.90	39.60	32.40
Ash (% of DM)	5.31	6.22	7.07
NDF (% of DM)	31.21	31.19	32.16
ADF (% of DM)	20.02	21.97	23.65
CP (% of DM)	14.30	14.80	14.90
EE (% of DM)	3.21	3.03	2.90
Lignin (% DM)	2.49	2.56	2.56
NEm ^3^ (Mcal/kg DM)	1.76	1.71	1.68
NEg ^3^ (Mcal/kg DM)	1.13	1.10	1.07

^1^: Premix, calcium carbonate 69 (g/day); potassium iodide 0.012 (g/day); potassium chloride 10 (g/day); sodium chloride 30 (g/day); sodium selenite 0.0949 (g/day); copper sulfate 1.03 (g/day); zinc sulfate 0.50 (g/day); manganese sulfate 0.10 (g/day); vitamin A 15,000 (UI/day); vitamin D 2250 (UI/d); yeast (Celmanax) 6 (g/day); and monensin al 20% 1.25 (g/day). ^2^: Calculated based on the National Academies of Sciences, Engineering and Medicine (NASEM, 2016). ^3^: Calculated according to Weiss et al. [22]. FCP: fresh citrus pulp. DM: dry matter. NDF: neutral detergent fiber. ADF: acid detergent fiber. CP: crude protein. EE: ether extract.

**Table 2 animals-11-03338-t002:** Least square means ± standard errors of animal performance according to dietary treatments.

	Dietary Treatments				*p*-Value		
Item	FCP0	FCP15	FCP30	Diet	Linear	Quadratic	FCP0 vs. FCP15 + 30
Initial weight (kg)	386.4 ± 7.97	385.1 ± 7.97	382.5 ± 8.19	0.8841	0.6278	0.9277	0.7015
Final weight (kg)	523.5 ± 10.6	520.1 ± 10.6	509.8 ± 11.0	0.4601	0.2328	0.7203	0.3761
ADG (kg BW/day) ^1^	1.580 ± 0.06	1.535 ± 0.06	1.480 ± 0.06	0.3629	0.1594	0.9260	0.2293
Dry matter intake (kg DM/day)	11.80 ^a^ ± 0.33	11.28 ^a^ ± 0.33	10.31 ^b^ ± 0.35	0.0066	0.0019	0.5551	0.0101
F:G (kg DM/kg BW) ^2^	7.51 ^a^ ± 0.15	7.37 ^a^ ± 0.15	6.98 ^b^ ± 0.16	0.0352	0.0168	0.4653	0.0699
Water intake (lt/day)	24.02 ± 0.79	23.20 ± 0.79	22.30 ± 0.82	0.4350	0.2309	0.6802	0.2084
WI:DMI (lt/kg DM) ^3^	2.04 ± 0.79	2.07 ± 0.79	2.25 ± 0.82	0.0500	0.0231	0.3054	0.1169

^1^: ADG, average daily gain (kg/day); ^2^: F:G, (average daily feed intake (kg DM/day)/average daily gain (kg BW/day)); ^3^: WI:DMI, (water intake (lt/day)/dry matter intake (kg/day)). ^a,b^: Values with different superscripts in the same row differ significantly (*p* < 0.05).

**Table 3 animals-11-03338-t003:** Least square means ± standard errors of carcass and meat quality traits according to dietary treatments.

	Dietary Treatments				*p*-Value		
Item	FCP0	FCP15	FCP30	Diet	Linear	Quadratic	FCP0 vs. FCP15 + 30
Hot carcass weight (kg) ^2^	274.4 ± 2.1	271.6 ± 2.1	269.9 ± 2.2	0.3463	0.1546	0.8403	0.1732
Marbling ^1,3^	407 ± 11.9	387 ± 11.7	397 ± 12.5	0.5113	0.5975	0.3144	0.3377
Ribeye area ^3^ (cm^2^)	63.2 ± 2.1	63.4 ± 2.1	63.1 ± 2.3	0.9941	0.9820	0.9156	0.9748
Fat thickness ^3^ (mm)	14.6 ± 1.2	12.7 ± 1.2	12.9 ± 1.3	0.4521	0.3407	0.4347	0.2221
WBSF–5 day	3.30 ± 0.23	3.11 ± 0.23	3.60 ± 0.25	0.3621	0.3920	0.2305	0.8482
Cooking losses (%)	22.5 ± 0.6	21.4 ± 0.6	21.9 ± 0.6	0.4058	0.4635	0.2626	0.2383
L*–0 day	38.9 ± 0.9	38.9 ± 0.9	38.8 ± 1.0	0.9911	0.9323	0.9172	0.9817
a*–0 day	22.4 ± 0.5	22.3 ± 0.5	21.6 ± 0.5	0.4233	0.2420	0.5521	0.4668
b*–0 day	11.8 ± 0.4	12.1 ± 0.4	11.4 ± 0.4	0.4103	0.4318	0.2769	0.8970
L*–5 day	39.8 ± 0.7	39.2 ± 0.7	38.3 ± 0.7	0.3672	0.1627	0.8843	0.2428
a*–5 day	22.2 ± 0.7	22.7 ± 0.7	22.0 ± 0.7	0.7415	0.7659	0.4749	0.9231
b*–5 day	10.6 ± 0.4	11.3 ± 0.4	10.4 ± 0.5	0.3476	0.7479	0.1580	0.6608

^1^: USDA marbling scores were encoded as follows: slight = 300 to 399, small = 400 to 499. ^2^: Adjusted by slaughter weight. ^3^: Adjusted by hot carcass weight. L*: lightness, a*: redness and b*: yellowness.

**Table 4 animals-11-03338-t004:** Least square means ± standard errors of micronutrient content and meat antioxidant capacity according to dietary treatments.

	Dietary Treatments				*p*-Value		
Item	FCP0	FCP15	FCP30	Diet	Linear	Quadratic	FCP0 vs. FCP15 + 30
α-tocopherol ^1^	0.96 ^c^ ± 0.32	2.26 ^b^ ± 0.34	3.54 ^a^ ± 0.34	<0.0001	<0.0001	0.9722	<0.0001
Retinol ^1^	0.17 ± 0.03	0.15 ± 0.03	0.18 ± 0.03	0.8413	0.7693	0.6091	0.9885
Retinoid ^1^	0.88 ^b^ ± 0.06	1.04 ^a,b^ ± 0.06	1.18 ^a^ ± 0.06	0.0044	0.0011	0.9096	0.0030
β-carotene ^1^	1.38 ^a,b^ ± 0.08	1.53 ^a^ ± 0.08	1.17 ^b^ ± 0.08	0.0084	0.0585	0.0086	0.7564
Carotenoids ^1^	2.99 ± 0.17	3.33 ± 0.18	2.83 ± 0.18	0.1506	0.5164	0.0649	0.6535
DPPH ^2^	125.4 ± 2.6	117.9 ± 4.0	119.3 ± 2.6	0.1604	0.1899	0.3161	0.0563
ORAC hydrophilic ^2^	414.6 ± 11.6	381.6 ± 11.6	393.3 ± 12.1	0.0999	0.1745	0.0975	0.0472
ORAC lipophilic ^2^	91.9 ^b^ ± 4.7	113.8 ^a^ ± 4.5	95.0 ^b^ ± 4.7	0.0022	0.6065	0.0006	0.0241

^1^: expressed as µg of each standard/g of muscle (wet basis). ^2^: expressed as µmol TE/100 g fresh weight. ^a,b^: Values with different superscripts in the same row differ significantly (*p* < 0.05).

## Data Availability

Data will be made available upon reasonable request.

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
