# Peer review of "Effect of Fresh Citrus Pulp Supplementation on Animal Performance and Meat Quality of Feedlot Steers"

_animals, 2021, doi:10.3390/ani11123338_

Round 1

Reviewer 1 Report

Dear authors,

Thank you very much for the paper,

I have some major questions.

Kind regards,

Reviewer

SPECIFIC COMMENTS

Replace throughout the document ºC by °C.

Replace throughout the document chromameter by colorimeter.

Introduction: In the introduction there should be a short paragraph talking about the comparison between silage and citrus pulp.

Line 48: Is citrus pulp rich in Protein?  And, for example, phosphorus?

Materials and Methods: As citrus pulp is rich in sugars and pectins, it is rapidly degraded in the rumen. In view of this fact, there was some disturbance by acidosis? Why was no sensory analysis carried out on the meat ?

Lines 148-150: Japan City is missing. Why did the authors use illuminant C instead of illuminant D65? Why have the coordinates C* and H° not been quantified?

Line 305: p

Table 5: I understand that the authors put the letters of the significant differences above the standard error. However, the significant differences are relative to the means. Therefore, I believe that the letters of the significant differences should appear above the mean values.

Lines 357-359: What would be the values/concentrations needed to affect colour quality? Authors [42] have observed higher values for L* and how does this influence the consumer? Is the influence positive or negative with higher L* values?

Author Response

Thank you to review our manuscript. Our responses are in the attached file.

Reviewer 2 Report

The manuscript titled "Effect of different levels of citrus pulp on animal performance and meat quality of feedlot steers" has been reviewed. The topic is not new and the experiment was not well designed, and the language is hard to follow without academic statement. Below are my detailed comments:

Line 1-Line 2: Effect of different levels of citrus pulp on animal performance and meat quality of feedlot steers. It seems "different levels of" is redundant, instead "supplementation" is needed in the title.

Line18-20: The use of crop, fruit and vegetable processing co-products for animal feeding has been of increasing interest worldwide to minimize feed waste which has a positive impact from the environmental standpoint. This sentence is so hard to follow, please rephrase it into two or more sentences.

Line 20-21: The use of many co-products has some limitations related to poor palatability for animals, logistic aspects to deliver the product. What " logistic aspects" here means?

Line 26-28:The findings of the study showed that citrus pulp may be used as an energy source in rations for feedlot steers not affecting animal performance, carcass, and meat quality but with a positive effect on FI and better FCR.  How can you conclude citrus pulp could be used as an energy source since no energy parameters were provided in diet and results? Besides, what is the meaning of FI and FCR when they are appeared for the first time?

Line 34-37: the inclusion of citrus pulp up to 30% of the diet did not affect the animal average daily gain but those steers consumed less feed and were more efficient converting feed into weight gain than their counterparts without citrus pulp in the diet. "were more efficient converting feed into weight gain than their counterparts" how to understand? The biggest problem for the abstract is no exact result is provided with only summative results.

Line 44-45: In addition, the use of fruit wastes for animal feeding would reduce costs incurred by farmers [3]. How to follow "incurred by farmers"?

Line 49-51: Animal performance of steers fed citrus pulp in the ration has been similar to those fed with corn diets [6] being suitable for its inclusion in a balanced diet replacing other energy feeds [7]. These findings indicate the current work is not new and I think more results could be found regarding CP application.

Line 76-77: the steers of the second treatment were offered a diet as treatment 1 but with 15% of citrus pulp (CP15) on dry matter basis. Why here should be a but?

Line 79: Citrus pulp was mainly composed of lemon. A more detailed composition should be provided here including active ingredients.

Table 1: Crude protein and metabolizable energy at least should be provided in part of Chemical Composition. With many differences in DM, NDF, ADF and NE, how can you conclude which finally determine the output? This is the biggest concern.

Table 2: With n=12, how can the initial weight with so much differences? Besides, WI:FI (Its/kg DM), what is the Its  means? Similar mistakes should be paid more attention.

Author Response

(The authors gave the same response as above.)

Reviewer 3 Report

Overall, this is a well written article. A few general suggestions would be to NOT use CP for your citrus pulp abbreviation. CP is the common abbreviation for crude protein. This led to difficulty reading your paper due to this abbreviation. Also suggest using dry matter intake (DMI) instead of FI and for the feed conversion ratio, the common abbreviation is F:G.

Line 28: no abbreviations; feed intake or feed conversion ratio, is not common use, suggest changing to dry matter intake and F:G

Abstract: p-values should be included and a brief conclusion statement. If you need room you can remove the sentence form line 30 to 32.

Line 67: remove "located...km 288"

Line 181: define BHT

Lines 249-252: orthogonal contrasts would be limited to 2, treatments (n = 3) - 1 = 2 contrast statements. You cannot run linear, quadratic, and citrus pulp vs. no citrus pulp. When presenting contrasts, you don't discuss or present the overall treatment effects, you only present the contrasts.

All results: only contrast results should be presented. All p-values should be uppercase, italicized P.

Tables 2 to 5: remove +/- and include a single standard error column to present SE. 2 decimals for p-values is sufficient

Figure 1: indicate on the graph when the feedings occurred

The discussion was well written.

Author Response

(The authors gave the same response as above.)

Reviewer 4 Report

The manuscript entitled ‘Effect of different levels of citrus pulp on animal performance and meat quality of feedlot steers’ studies the use of the by-product citrus pulp as an energy source in rations for feedlot steers. However and primarily, the novelty of the study is necessary to be clarified and emphasized, since there have been already published a lot of studies performed in lambs or cattle.

In introduction, as there are many previous studies in ruminants you must present the commonly used levels of citrus pulp in the rations to avoid any potential negative effects and in order to explain the reason that you chose the two studied levels of CP. Also, you should write in Materials and Methods if you fed the animals with dried or fresh citrus pulp (you refer it for the first time in Conclusion).

The hypothesis, in lines 59-61 - ‘…would not have detrimental effects on animal performance but having a positive impact on meat quality’ – is based on the results of the study. Please rephrase it or delete it.

In line 204: What kind of dried samples? Were they LT dried samples?

When you use a reference writing the phrase ‘described by’ (e.g. in lines 178 and 216) you should add the first author’s name in the text

In the paragraph 319-325, the interpretation of the result (lines 323-325) does not seem scientific. Instead, you may consider the higher water holding capacity of citrus pulp (see reference 4) which could probably contributed to the higher WI:FI ratio found.

Author Response

(The authors gave the same response as above.)

Round 2

Reviewer 1 Report

Dear Authors,

Congratulations on your manuscript,

Kind Regards

Reviewer

Author Response

Thank you very much for your comments.

No new edits were asked by the Reviewer 1.

Reviewer 2 Report

After re-reviewing the manuscript "Effect of fresh citrus pulp supplementation on animal perfor-mance and meat quality of feedlot steers", I feel sorry to maintain the previuos decision because of the response. The reasons are listed as follows:

L1: Why should citrus pulp fed in a fresh way?

L80: Chemical composition of the citrus pulp should be provided, since you have indicated "fresh citrus pulp may be used as an energy source in rations' in L27.

L83-88: Why you choose the dose of 15% and 30% of fresh citrus pulp on dry matter basis? This must be well explained.

Figure 1: The image is too blurry.

Table 5: Why should here be different from the previous version regarding data?

Author Response

Thank you for your comments to improve the manuscript.

Please see the attached Word file with our responses.

Reviewer 4 Report

The recommended revisions were incorporated into the manuscript.

Author Response

The suggestions/edits were included in the previous version.

No new edits were asked by Reviewer 4.

Round 3

Reviewer 2 Report

After reviewing your 3rd revisions, I still feel not the current form to publish. However, it matters nothing if editor or other reviews permit acceptance. Below is my concerns:

L44 Introduce: some recent publications (2020-2021) have discussed the utilization of fruit wastes or residue, but authors just have mentioned some of them. Please try to improve it.

L80: Chemical composition of the citrus pulp should be provided, since you have indicated "fresh citrus pulp may be used as an energy source in rations' in L27.

That statement in the Summary section (L. 27) is based on the Metabolized Energy (ME) of citrus pulp that was 2.54 Mcal/kg DM. We included in the Materials and Methods section the ME and CP data of citrus pulp (Line 88-89).

Comment: where can I find the ME in Line 27.

L83-88: Why you choose the dose of 15% and 30% of fresh citrus pulp on dry matter basis? This must be well explained.

We needed to evaluate three supplementation levels (including 0%) to study the response (response curve) of the inclusion of citrus pulp in the diet. The dose of 30% of citrus pulp was based on previous research were the authors reported that high levels of citrus pulp supplementation had negative impacts on animal performance (even rumen parakeratosis reported in lambs). In addition, a dose above 30% of fresh citrus pulp would not only increase water in the diet but also transport costs from the citrus industry to the farm.

Comment: The reason to choose the dose of 15% should be well explained in text, not only in response, and the every statement in experiment design should provide related reference.

The font type in figure is not in accordance to the format requirements.

L450-451:Kasapidou, E.; Sossidou, E; Mitlianga, P. Fruit and Vegetable Co-Products as Functional Feed Ingredients in Farm Animal Nu-450 trition for Improved Product Quality. Agriculture 2015, 5, 1020–1034, https://doi.org/10.3390/agriculture5041020 Here, the title is not impressed in a right by apitalizing every word. Three rounds revisions have been made, but these small mistakes still remain.
